# Establishment of a congenic strain for the oyster mushroom reveals the structure and evolution of mating-type loci

Yi-Yun Lee[1], Guillermo Vidal-Diez de Ulzurrun [1], Rebecca J. Tay [2], Yen-Ping Hsueh [1,2]*

**1** Department of Complex Biological Interactions, Max Planck Institute for Biology, Tübingen, Germany,
**2** Institute of Molecular Biology, Academia Sinica, Nangang, Taipei, Taiwan

* yphsueh@tuebingen.mpg.de

## Abstract

*Pleurotus ostreatus*, a widely cultivated edible oyster mushroom, is an ecologically versatile species with applications in biotechnology, agriculture, and food production. It functions as a decomposer and in nutrient-limited conditions it enhances its survival by using a potent toxin to prey on nematodes. Its adaptability is further regulated by sexual reproduction, which follows a tetrapolar mating system governed by two unlinked, multiallelic loci, *matA* and *matB*. The two mating-compatible monokaryotic strains PC9 and PC15, derived from the parental dikaryon strain N001, exhibit significant physiological differences. PC9 grows robustly in laboratory conditions, whereas PC15 grows more slowly, making PC9 the preferred strain for research. To advance *P. ostreatus* as a genetic model, we characterized the mating-type (*MAT*) loci of both monokaryon strains and developed a congenic strain. We analyzed the *MAT* loci in multiple *P. ostreatus* strains, and identified 11 A and 12 B alleles among twelve haplotypes, confirming their multiallelic nature. Using 10 rounds of backcrossing, we introgressed the *matA* and *matB* loci from PC15 into the PC9 genetic background to generate the congenic strain PC9.15. After sequencing and assembling a high-quality and contiguous genome for PC9.15, we confirmed that the genomes of PC9.15 and PC9 are 99% similar, with the only major difference placed at the *matA* and *matB* loci.

## Author summary

Mushrooms are not just decomposers-they can also be predators. The oyster mushroom *Pleurotus ostreatus* is an ecologically adaptable species that thrives by breaking down plant material and preying on nematodes using a potent toxin that triggers rapid paralysis of nematode prey. In addition to its ability to switch from saprophyte to predator, another feature contributing to the ecological success of *Pleurotus* is its reproductive strategy. *P. ostreatus* reproduces sexually through a tetrapolar mating-type system, governed by two unlinked mating loci,

**Data availability statement:** The genomic data has been deposited to the National Center for Biotechnology Information GenBank under the accession numbers JAQZRA000000000.1 (PC9.15), OR030114 (PC9 mitochondrial genome), JBQVBD000000000 (SS5), and JBQVBE000000000 (SS14). The accession numbers of BioProject are PRJNA1098946 (SS5) and PRJNA1098945 (SS14). The raw Nanopore reads of P. ostreatus PC9.15, SS5, and SS14 have been uploaded to the Sequence Read Archive (SRA) and are available under the accession numbers SRR23581323, SRR28975996, and SRR28975997, respectively. https://www.ncbi.nlm.nih.gov/nuccore/JAQZRA000000000.1/ https://www.ncbi.nlm.nih.gov/nuccore/OR030114 https://www.ncbi.nlm.nih.gov/nuccore/JBQVBE000000000.1/ https://www.ncbi.nlm.nih.gov/bioproject/PRJNA1098946/ https://www.ncbi.nlm.nih.gov/bioproject/?term=PRJNA1098945 https://www.ncbi.nlm.nih.gov/sra/?term=SRR23581323 https://www.ncbi.nlm.nih.gov/sra/?term=SRR28975996 https://www.ncbi.nlm.nih.gov/sra/?term=SRR28975997.

**Funding:** This work was supported by the Max Planck Society to Y.-P.H., Academia Sinica Investigator Award AS-IA-111-L02 to Y.-P.H., and NSTC grant 110-2311-B-001-047-MY3 to Y.-P.H. The funders had no role in study design, data collection and analysis, decision to publish, or preparation of the manuscript.

**Competing interests:** The authors have declared that no competing interests exist.

*matA* and *matB*. These loci determine mating compatibility and contribute to genetic diversity, yet their precise structure and allelic diversity remain unclear. In this study, we characterized the *matA* and *matB* loci in *P. ostreatus* strains PC9 and PC15 and analyzed *MAT* loci across multiple strains, confirming their multiallelic nature. To enhance *P. ostreatus* as a genetic model, we developed the congenic strain PC9.15 by introgressing *MAT* alleles from PC15 into the PC9 genetic background through genetic backcrosses. Genome sequencing confirmed that PC9.15 closely matches the genome of PC9, except at the *MAT* loci. This study expands our understanding of mating-type diversity in *P. ostreatus*, introduces PC9.15 as a valuable tool for genetic studies and controlled crosses, and facilitates the development of *P. ostreatus* as a model system for fungal biology.

## Introduction

The edible oyster mushroom *P. ostreatus* is a fascinating organism with significant ecological and industrial importance. It is one of the most widely cultivated edible mushrooms worldwide and an ecologically versatile species, thriving as a decomposer in forests on decaying wood and forest litter [1,2]. Nutrient-limited environments can trigger a shift in *P. ostreatus* toward carnivory to supplement its nutrient intake. *P. ostreatus* preys on nematodes by producing a potent toxin, 3-octanone, which is concentrated and stored inside specialized structures called toxocysts. Upon release, this toxin rapidly paralyzes nematodes within minutes [3–5], allowing the fungus to consume the immobilized prey. This ability to exploit multiple nutrient sources using two different lifestyles contributes to *Pleurotus*'s adaptability and ecological success.

Like many fungi, *P. ostreatus* relies on sexual reproduction to maintain genetic diversity and adaptability [2]. In fungi, sexual reproduction is regulated by mating-type loci (*MAT*), which determine compatibility and control genetic exchange between individuals [6]. Fungal mating systems are classified into heterothallism, where individuals require a genetically compatible partner, or homothallism, which is characterized by self-fertilization [7]. The *MAT* loci of fungi have evolved into two different organization paradigms: the bipolar and tetrapolar systems [8]. In bipolar mating systems, the homeodomain (*HD*) and pheromone/pheromone receptor (*P/R*) genes are linked at a single *MAT* locus which determines sexual compatibility, requiring two isolates to carry opposite alleles at this locus for successful mating [6]. Conversely, tetrapolar mating systems involve two unlinked *MAT* loci: the *HD* locus, encoding two homeodomain-type transcription factors, and the *P/R* locus, which contains tightly linked mating pheromones and pheromone receptors; mating in tetrapolar systems requires distinct alleles at both loci for compatibility [6]. Among Basidiomycetes, the majority of species retain this ancestral tetrapolar mating system. However, in certain lineages such as *Cryptococcus neoformans* and *Ustilago hordei*, the two loci have fused, resulting in a derived bipolar mating system [9,10]. *P. ostreatus* follows a heterothallic bifactorial tetrapolar mating system, where sexual compatibility is

determined by two unlinked, multiallelic loci, *matA* and *matB* [11,12]. Analyses of 17 *P. ostreatus* strains have revealed at least ten distinct *matA* alleles and eight *matB* alleles, confirming the extensive allelic diversity at both loci [13]. Two widely used monokaryotic strains, PC9 and PC15, have been derived from the parental dikaryotic strain N001 through a process that isolates distinct haploid nuclei called dedikaryotization [14]. PC9 carries the mating type *A2B1*, while PC15 harbors *A1B2* [14]. Although the mating-type loci of *P. ostreatus* have been described previously [13], their detailed structural organization and allelic diversity remain incompletely characterized. A deeper understanding of these loci is crucial for elucidating the molecular basis of mating compatibility and identifying potential structural variations between strains.

Congenic strains serve as invaluable tools for genetic studies, retaining the genetic identity of a parental strain while incorporating a specific region of interest. Such strains enable precise investigations into complex genetic pathways and have been used to study the function of mating-type loci in sexual and unisexual reproduction, virulence, uniparental mitochondrial inheritance, and enable efficient genetic mapping and bulked segregant analysis (BSA) in various fungal models [15–17]. Despite their broad utility, congenic strains are not available in *P. ostreatus*, limiting efficient genetic analyses and manipulation in this species. A clear need for congenic strains in *P. ostreatus* emerges from a genomic comparison between two commonly used laboratory strains, PC9 and PC15. The PC9 and PC15 genomes align with approximately 97% identity over 94% of the genome, with the remaining 6% showing no alignment [18]. This high level of genetic variation likely contributes to two strains' differing physiological characteristics: PC9 grows at least twice as fast as PC15 in laboratory conditions, highlighting the significant impact of their genetic differences and making PC9 the preferred laboratory reference strain [18].

Here, we report the characterization of the *MAT* loci and the development of a congenic strain of *P. ostreatus*, PC9.15, through ten genetic backcrosses. We sequenced and assembled the PC9.15 genome using a long-read (Nanopore) sequencing, enabling comparison of the *MAT* loci and their flanking regions between PC9 and PC9.15. We further compared the *MAT* loci among multiple *P. ostreatus* genomes and revealed its multiallelic structure. From twelve haplotypes analyzed, we identified 11 distinct alleles at *matA* and 12 at *matB*, highlighting the remarkable genetic diversity of the *MAT* loci in the *P. ostreatus* population. The development of the congenic strain, PC9.15, in the genetic background of the model strain PC9, enhances the genetic toolkit for *P. ostreatus*, offering a powerful genetic tool and resource to advance research in mushroom biology.

## Results

### Characterization of the mating-type loci in *P. ostreatus* strains PC9 and PC15

High-quality genome assemblies of *P. ostreatus* strains PC9 and PC15 are available [18–21], yet the mating-type loci remain incompletely characterized. To improve gene model accuracy within the mating-type regions and to ensure consistent locus characterization between strains, we reannotated the PC15 genome available from JGI (assembly accession number: GCA_000697685.1). In basidiomycetes, the *matA* locus is typically flanked by the highly conserved mitochondrial intermediate peptidase (*MIP*) and β-flanking (*βFG*) genes [22]. Using these conserved flanking genes, we identified and characterized the boundaries of the *P. ostreatus matA* locus. The locus spans ~5kb in PC9 and ~10kb in PC15; both are located on chromosome 2 (Fig 1A). The *matA* locus encodes two homeodomain (*HD*) genes, *HD1* and *HD2*, which can form heterodimers with compatible HD proteins from the opposite mating partner, enabling intracellular recognition of sexual compatibility after mating [23]. Notably, PC9 carries two HD genes (*HD1.1* and *HD2.1*) transcribed in opposite directions, whereas, PC15 harbors four *HD* genes (*HD1.2*, *HD1.3*, *HD2.2*, and *HD2.3*) in the *HD* locus (Fig 1A), suggesting a gene duplication event in PC15. A previous study reported only a single *HD2* gene in the PC15 *matA* locus [13]; however, examination of the PC15 JGI genome assembly (GCA_000697685.1) confirmed the presence of two *HD2* genes [19,21,20], consistent with the complete gene content observed in our analysis. To assess allelic divergence, we aligned the protein sequences of HD1 and HD2 from both strains. While all HD proteins share conserved homeodomain motifs,

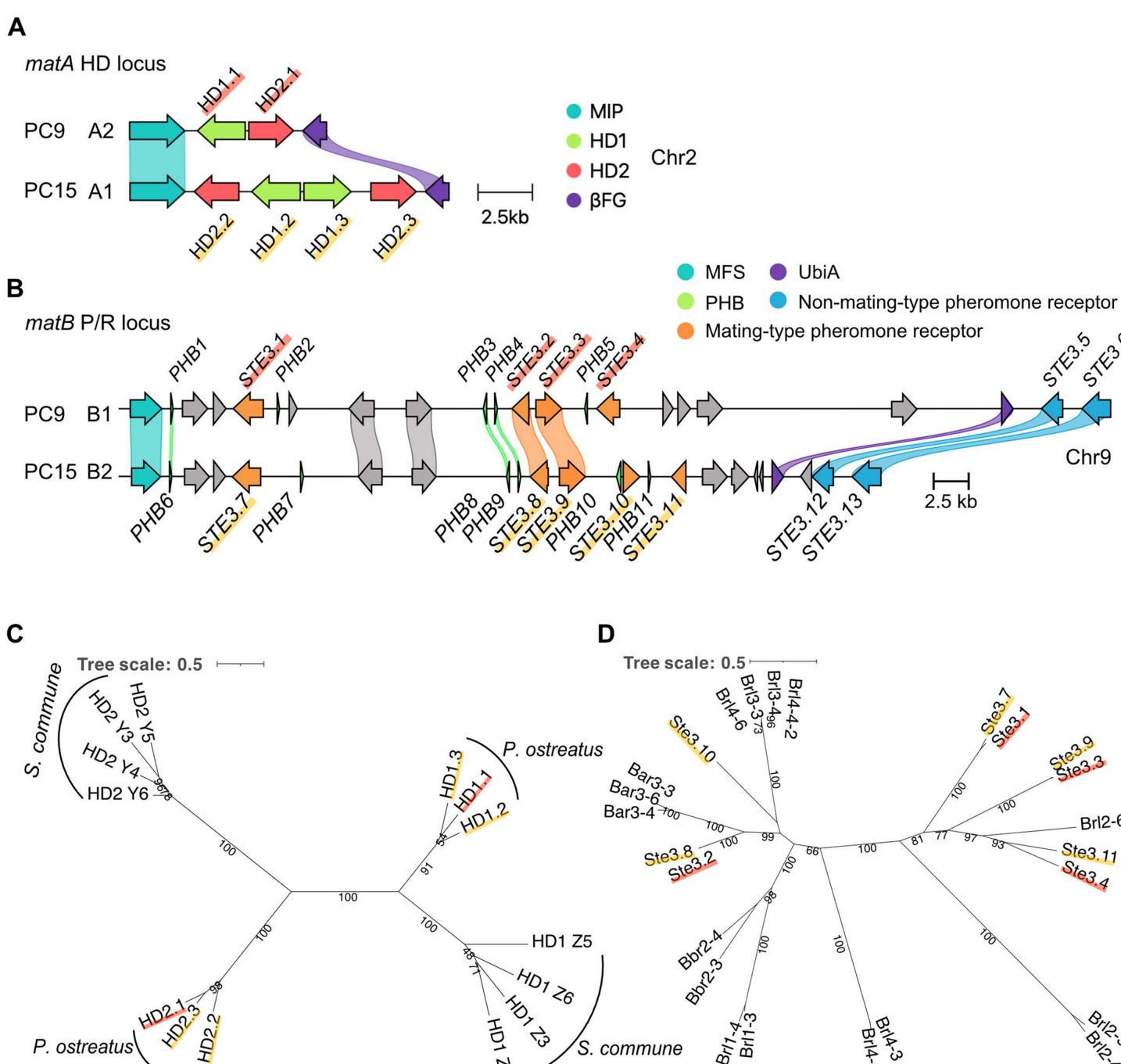

**Fig 1. Characterization of mating-type loci and associated genes in *P. ostreatus*. (A and B)** *MAT* locus alignments (amino acid-based) of strains PC9 and PC15 (clinker v0.0.31). Genes encoding proteins having more than 95% identical amino acid residues are linked. The color code for links and genes highlights sequence homology and grey genes have no similarity to other genes in the alignment. **(A)** In the *matA* locus on chromosome 2 (Chr2), homeodomain genes *HD1* and *HD2* are shown in green and pink, respectively; the conserved mitochondrial intermediate peptidase (*MIP*) and beta flanking gene (*βFG*) are shown in teal and purple, respectively. **(B)** In the *matB* locus on chromosome 9 (Chr9), mating-type specific pheromone receptor genes (*STE3*) are shown in orange (within *matB*) and blue (outside *matB*); pheromone genes (*PHB*) are in green; the conserved major facilitator superfamily (*MFS*) transporter and UbiA family prenyltransferase (*UbiA*) are shown in teal and purple, respectively; other genes are shown in grey. Scale bars indicate 2.5 kb. **(C and D)** Phylogenetic trees of HD proteins **(C)** and Ste3 proteins **(D)** from *P. ostreatus* (PC9 in red underline; PC15 in yellow underline) and *S. commune* by the maximum-likelihood method using IQ-TREE.

we still observed distinct sequence variations among HD1 and HD2 protein (S1A and S1B Fig), supporting the conclusion that PC9 and PC15 possess genetically distinct and mating-compatible *matA* alleles.

In both PC9 and PC15, the *matB* locus is located on chromosome 9, spanning ~54 kb and ~40 kb, respectively (Fig 1B). The locus contains mating pheromone precursor (*PHB*) and pheromone receptor (*STE3*) genes, which are involved in premating recognition [6]. In the model basidiomycetes, the *matB* locus is flanked by various genes. For example, in *C. cinereus,* the *B* locus is bordered by a gene encoding major facilitator superfamily (*MFS*) transporter [24,25], whereas in *S. commune*, the *matB* locus is flanked by genes encoding a zinc finger transcription factor and hypothetical proteins [26]. Similarly, in *P. ostreatus* strain PC9 and PC15, the *matB* is flanked by an *MFS* transporter on one side and a prenyltransferase (*UbiA*) gene on the other, providing genomic boundaries for *matB* locus. To identify the mating-type genes within *matB*, we searched for conserved Pfam domains corresponding to *STE3* (Pfam: PF02076) and *PHB* (Pfam: PF08015) domains. In total, four *STE3* genes (*STE3.1* to *STE3.4*) and five *PHB* genes (*PHB1* to *PHB5*) were identified in PC9, and five *STE3* genes (*STE3.7* to *STE3.11*) and six *PHB* genes (*PHB6* to *PHB11*) were identified in PC15 (S1 Table). Four additional genes encoding Ste3 proteins, *STE3.5*, *STE3.6*, *STE3.12*, and *STE3.13*, that share high sequence identity (> 96%) were found flanking the *matB* locus, suggesting that they are not mating-type-specific (S2 Table). Alignment of all nine *STE3* receptors encoded in the *matB* locus revealed notable variation in the C-terminal regions (S1C Fig), consistent with their roles in mating signal recognition and allelic specificity. We further identified genes that encode mating pheromone precursors (*PHB*) within the *matB* locus. Across both alleles, we identified 11 *PHB* genes encoding peptides of 51–72 amino acids in length with a canonical C-terminal CAAX motif (C = cysteine, A = aliphatic residue, X = any residue) characteristic of mating pheromones [27]. Protein sequence alignment confirmed conservation of this motif among all *P. ostreatus* Phb peptides (S1E Fig).

To explore the evolutionary divergence of mating-type genes across basidiomycetes, we constructed maximum-likelihood phylogenetic trees of HD and Ste3 protein from *P. ostreatus* and the model mushroom *Schizophyllum commune* (Fig 1C and 1D). Phylogenetic analysis of HD proteins revealed two distinct, well-supported clades corresponding to HD1 and HD2 (bootstrap support = 100%). Within each clade, sequences clustered by species, forming subgroups indicative of taxon-specific diversification (Fig 1C). The clear separation between HD1 and HD2, along with their internal species-level groupings, suggests long-term evolutionary maintenance and potential functional specialization of these mating-type genes. In contrast, Ste3 proteins grouped first by allele type and then further separated by species (Fig 1D). Within these, Ste3.2 and Ste3.8 shared 99.5% amino acid identity, and Ste3.3 and Ste3.9 shared 96.7% identity (S1D Fig and S2 Table). By contrast, the protein sequences of Ste3.1 versus Ste3.7 (75.7% identity) and Ste3.4 versus Ste3.11 (55.4% identity) are considerably more divergent, further supporting their roles as distinct mating-type pheromone receptors (S1D Fig).

## Multiallelic mating-type loci in *P. ostreatus*

In tetrapolar fungi, mating-type loci are often multiallelic rather than biallelic, resulting in the presence of hundreds or even thousands of mating-types, as seen in *C. cinereus* and *S. commune*, which have over 12,000 and 20,000 mating types, respectively [28]. *P. ostreatus* also follows a multiallelic tetrapolar mating system, though the exact structure and diversity of its mating-type alleles remain poorly characterized. To investigate its multiallelism, we compared the *matA* and *matB* loci across several *P. ostreatus* genomes (Table 1). Our dataset comprised two monokaryotic strains, SS5 and SS14, derived from strain TWF713, as well as additional publicly available *P. ostreatus* genomes retrieved from NCBI and JGI databases. These include monokaryotic strains CCMSSC00389 (389) [29], CCMSSC03989 (3989) [30], two haplotypes of the dikaryotic strain gfPleOstr1.1 (gf and gfalt) [31], and the dikaryotic strains DSM11191 (DSM) and 595 [32]. We then examined the *matA* and *matB* regions across all strains, focusing on their organization, gene content, and allelic variation to better understand the molecular basis of multiallelism in *P. ostreatus*. In this study, *matA* and *matB* alleles are defined based on complete nucleotide and protein identity. Alleles are distinct if they differ in nucleotide and protein sequence.

**Table 1. *P. ostreatus* genomes used in the present study.**

| Strain (haplotype) | Genome size | BUSCO (%) | Source | Assembly accession | Citation |
|---|---|---|---|---|---|
| PC9 | 35.0 Mb | 98.6% [S:97.3%, D:1.3%] | NCBI | GCA_014466165.1 | [18] |
| PC15 | 34.3 Mb | 99.3% [S:98.0%, D:1.2%] | NCBI | GCA_000697685.1 | [19–21] |
| SS5 | 41.4 Mb | 99.4% [S:95.9%, D:3.5%] | this study | GenBank WGS: JBQVBD000000000 | this study |
| SS14 | 43.6 Mb | 99.4% [S:91.2%, D:8.2%] | this study | GenBank WGS: JBQVBE000000000 | this study |
| CCMSSC00389 | 35.1 Mb | 98.8% [S:97.4%, D:1.4%] | NCBI | GCA_001956935.2 | [29] |
| CCMSSC03989 | 34.4 Mb | 98.9% [S:97.8%, D:1.1%] | NCBI | GCA_003313235.2 | [30] |
| gfPleOstr1.1 | 40.6 Mb | 99.4% [S:98.2%, D:1.2%] | NCBI | GCA_947034855.1 | [31] |
| gfPleOstr1.1 alternate haplotype | 40.8 Mb | 99.1% [S:97.4%, D:1.7%] | NCBI | GCA_947034875.1 | [31] |
| DSM11191 | 65.9 Mb | 99.5% [S:6.6%, D:92.9%] | JGI | JGI: PleosDSM 11191 v1.0 | |
| 595 | 79.7 Mb | 99.7% [S:25.7%, D:74.0%] | NCBI | GCA_024195665.1 | [32] |

Comparison of *HD* genes across ten *P. ostreatus* genomes revealed 17 *HD1* and 12 *HD2* alleles within these *matA* loci (Fig 2A and S3 Table). While at least 12 distinct *matA* alleles have been reported previously [11], their structural organization and sequence-level variation were not described. To avoid potential overlap with existing allele designations, we named the characterized alleles in this study from A21-A29. However, we cannot rule out the possibility that some of these may be identical in sequence or in mating type to previously reported alleles. SS14 harbors a *matA* allele identical to that of PC9 (allele A2), exhibiting complete sequence identity at both nucleotide and amino acid levels. Nine novel *matA* alleles were identified among the remaining haplotypes (Fig 2A). Six haplotypes contained a single *HD1*/*HD2* gene pair (Fig 2A), although duplication of *HD1* was observed in SS5 (A21), 389 (A22), and 595 (A26). Notably, SS5 (A21) carries three *HD1* genes, a configuration similar to that recently reported for *P. ostreatus* strain PC80, derived from an artificially cultivated strain HeiKang 650 [33]. The gfalt haplotype (A29) exhibited duplication of both *HD1* and *HD2*, consistent with our previous observations in PC15. Moreover, the *matA* locus of gfalt (A29) expanded to 73 kb—substantially larger than the 6 kb locus in PC9—and contained an insertion of five genes, including four hypothetical proteins and one transposase (Fig 2A). Given the sequence variation and gene duplications we had observed, we performed phylogenetic analysis to assess the divergence of HD1 and HD2 proteins across strains. This analysis revealed two distinct clusters for HD1 (HD1.1 to HD1.17) and HD2 (HD2.1 to HD2.13), confirming the divergence of these gene families (Fig 2C).

At the *matB* locus, we identified ten new alleles (B21-B30) that are absent in either PC9 or PC15, with locus sizes ranging from 45 kb to 150 kb (Fig 2B). The locus is flanked by *MFS* and *UbiA* genes, and most alleles contain three to five *STE3* and four to six *PHB* genes. One to two *STE3*-like genes were also found immediately outside the *UbiA* boundary in different strains; these receptors share high similarity and therefore were considered non-mating-type-specific. The *matB* loci of SS5 (B21), SS14 (B22), and 595 (B28) were notably expanded—to 105 kb, 77 kb and 150 kb, respectively—compared to the 54 kb locus in PC9. Several additional genes were detected within these expanded regions, including an integrase, a retrotransposon Gag protein, and hypothetical proteins, which are unlikely to be directly involved in mating but may have contributed to the locus expansion. Phylogenetic analysis of Ste3 protein sequences illustrates their diversity across *P. ostreatus* strains, aligning with the multiallelic nature of the *matB* locus and with the locus's role in facilitating diverse mating interactions (Fig 2D). In total, we identified distinct 12 *matB* distinct alleles across strains containing 48 *STE3*-like pheromone receptor genes, confirming the multiallelic configuration of both loci in *P. ostreatus*.

To test whether allelic variation affects mating compatibility, we performed reciprocal crosses among SS5 (*A21B21*), SS14 (*A2B22*), PC9 (*A2B1*) and PC15 (*A1B2*). SS5 is compatible with both PC9 and PC15, as evidenced by the presence of clamp connections—hallmark structures of dikaryotic hyphae in basidiomycetes—and successful fruiting body development (Fig 3). The SS14 x PC9 cross produced neither clamp connections nor fruiting bodies, as expected from their both harbor the *A2* allele of the *matA* locus (Fig 3). In contrast, the SS14 x PC15 cross yielded clamp connections

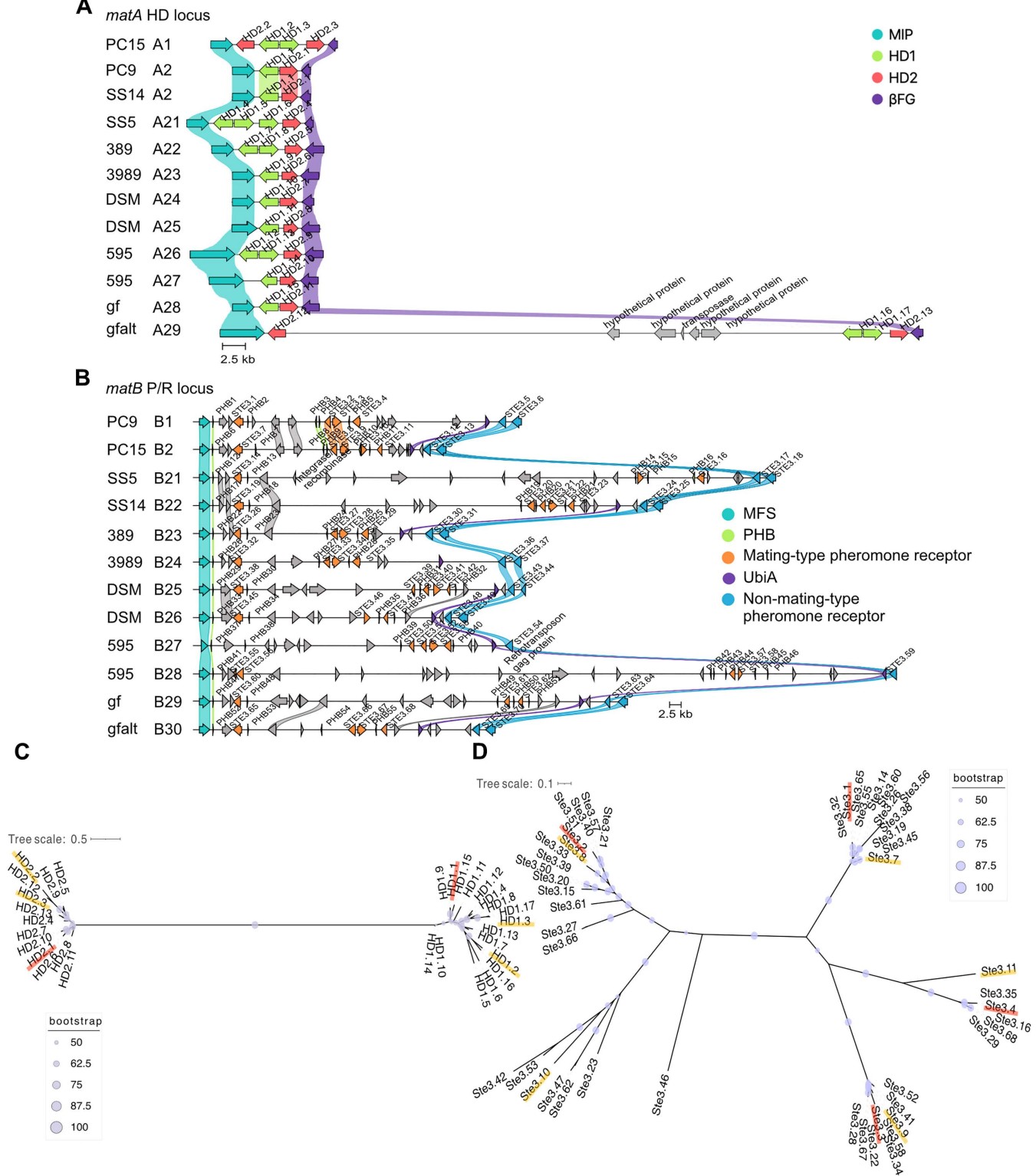

**Fig 2. Comparative analysis of *matA* and *matB* loci across *P. ostreatus* strains. (A and B)** *MAT* locus alignments (amino acid-based) of 12 haplotypes PC9, PC15, SS5, SS14, CCMSSC00389 (389), CCMSSC03989 (3989), gfPleOstr1.1 (haplotypes gf and gfalt), and two dikaryotic strains: DSM11191 (DSM) and 595 (clinker v0.0.31). Genes encoding proteins having more than 95% identical amino acid residues are linked. The color code for links and genes highlights sequence homology and grey genes have no similarity to other genes in the alignment. **(A)** Homeodomain genes *HD1* and

HD2 are shown in green and pink, respectively; the conserved *MIP* and *βFG* are shown in teal and purple, respectively. **(B)** Pheromone receptor genes (*STE3*) are shown in orange (within *matB*) and blue (outside *matB*); pheromone genes (*PHB*) are in green; the conserved *MFS* and *UbiA* are shown in teal and purple, respectively; other genes are shown in grey. **(C and D)** Phylogenetic trees of 30 HD proteins **(C)** and 48 Ste3 proteins **(D)** from 12 *P. ostreatus* haplotypes by the maximum-likelihood method using IQ-TREE (PC9 in red underline; PC15 in yellow underline).

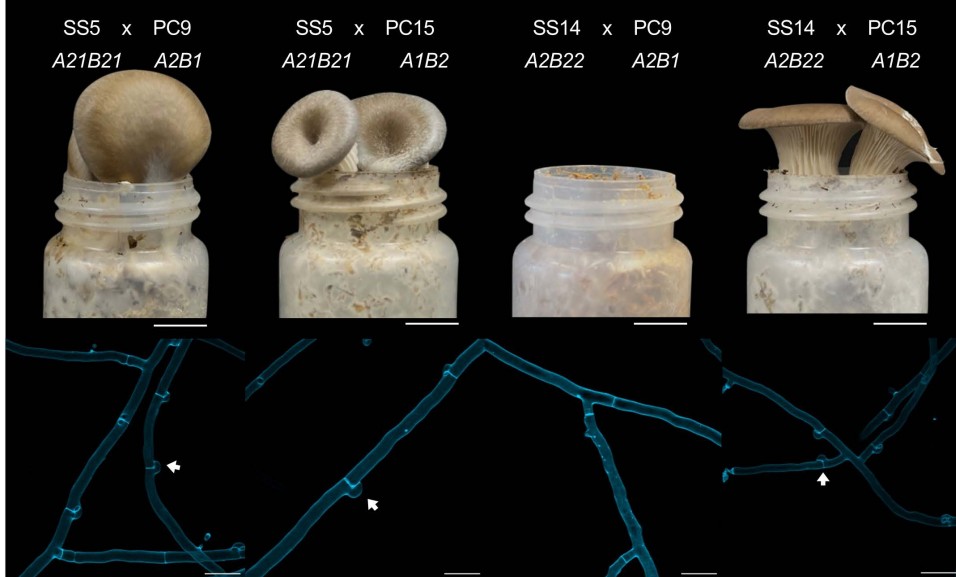

**Fig 3. Clamp connection and fruiting body formation in interstrain crosses of *P. ostreatus*.** Representative crosses of SS5 (*A21B21*) x PC9 (*A2B1*), SS5 (*A21B21*) x PC15 (*A1B2*), SS14 (*A2B22*) x PC9 (*A2B1*), and SS14 (*A2B22*) x PC15 (*A1B2*) display hallmark features of successful mating in a tetrapolar system, including clamp connections (arrows) and the fruiting body formation. Scale bar = and 1 cm (upper panels) and 10 μm (lower panels).

and fruiting bodies (Fig 3). These results support that *P. ostreatus* follows a tetrapolar mating system, in which successful mating requires compatibility at both *matA* and *matB* loci.

## Construction of the congenic strain PC9.15 via genetic backcrossing

PC9 and PC15 are monokaryotic strains derived from the dikaryotic strain N001, but only ~94% of their genomes are aligned with high similarity (~97%). This high genetic diversity likely contributes to the significant growth differences observed between the two strains. Given the robust growth and well-annotated genome of PC9, we selected the PC9 genetic background for introgression of the PC15 mating-type loci to generate a congenic strain (Fig 4A). To construct a congenic strain with PC9 background, we first crossed PC9 and PC15 and isolated meiotic progeny by dissecting individual basidiospores. Randomly selected progeny carrying the *A1B2* mating type as determined by PCR was then backcrossed with PC9 (*A2B1*). After nine additional rounds of backcrossing, we obtained a congenic strain that we designated PC9.15, which carries the *A1B2* mating-type loci from PC15 (Figs 4B and S2A). We found that PC9.15 exhibited growth behavior comparable to PC9 on nutrient-rich medium (YMG) and readily formed fruiting bodies when crossed with PC9 (Fig 4A and 4C). Overall, phenotypic characterization of PC9.15 showed that it retains growth and reproductive traits of PC9 while carrying the *A1B2* mating-type loci from PC15 (S2 Fig).

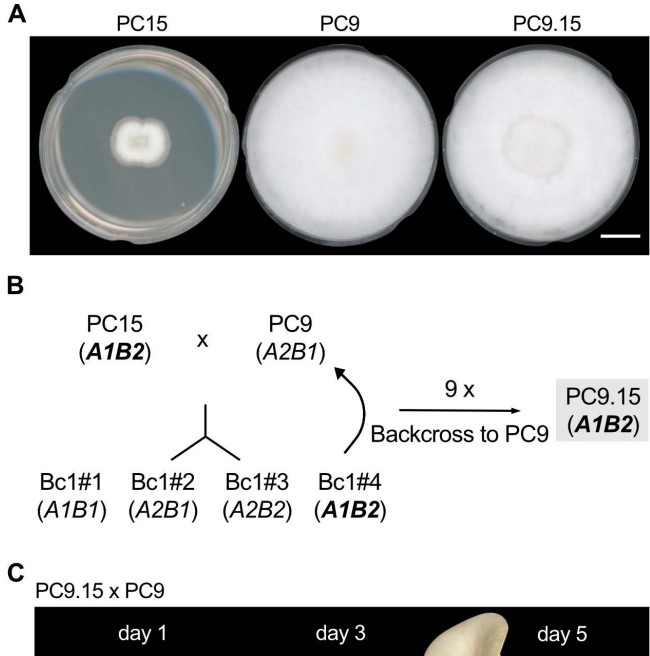

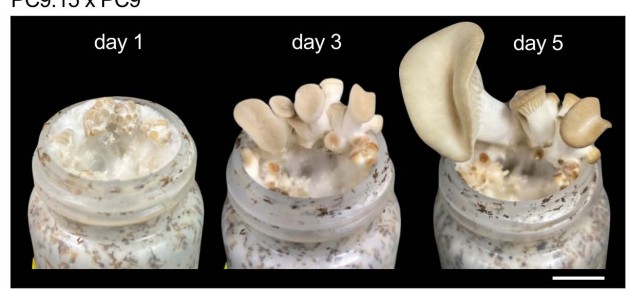

**Fig 4. Construction of the congenic strain PC9.15 with a PC9 genetic background and PC15 mating-type loci. (A)** PC9, PC15, and PC9.15 were grown on YMG agar media in 5.5-cm diameter Petri plates for 10 days. Scale bar = 1 cm. **(B)** Crossing strategy used to generate the PC9.15 congenic strain: Monokaryotic "Bc1#" progeny were obtained from a PC9 x PC15 cross. Progeny carrying the PC15 (*A1B2*) mating type were backcrossed to PC9 (*A2B1*) to generate monokaryotic "Bc2" progeny. This process was repeated for ten crosses to produce PC9.15. **(C)** Fruiting body development from a PC9.15 x PC9 cross at days 1, 3, and 5. Scale bars = 1 cm.

## High-quality genome assembly and analysis of the congenic strain PC9.15

To confirm the genome similarity of PC9.15 to PC9 and to characterize its chromosomal architecture, we generated a high-quality, near-chromosome-level genome assembly of the *P. ostreatus* congenic strain PC9.15. Using Nanopore sequencing, we achieved 128x coverage and assembled a contiguous genome of ~35.3 Mb (Table 2), comparable to the genome sizes of PC9 (35.0 Mb) and PC15 (34.3 Mb) [18–21]. The PC9.15 genome is distributed across 14 scaffolds, ranging from 38.7 kb to 5.36 Mb, including one scaffold representing the mitochondrial genome. Among the 11 main scaffolds analyzed, nine exhibit telomere-to-telomere assembly, one possesses telomeric repeats at a single end, and one lacks telomeric sequences. Although *P. ostreatus* is known to contain 11 chromosomes [14], our assembly of PC9.15 includes 13 scaffolds. The two additional scaffolds, representing only 1.4% of the total genome size, likely representing unresolved gaps or unplaced regions, such as centromeric or telomeric sequences. For example, scaffold 12 (468,928 bp) contains telomeric repeats and may correspond to a terminal chromosome region, while scaffold 13 (38,780 bp) shows high coverage depth but without gene density, a characteristic associated with centromeres [34], suggesting it may represent a misplaced centromeric fragment. Collectively, these results indicate that the PC9.15 genome assembly reaches near-chromosome-level resolution, with 11 chromosome-like scaffolds.

**Table 2. Genomic features of three *P. ostreatus* genome assemblies: PC9, PC9.15, and PC15. nt, nucleotides.**

| General features | PC9 | PC9.15 | PC15 |
|---|---|---|---|
| Total nt | 35,032,978 | 35,378,061 | 34,342,730 |
| Number of chromosome-like scaffolds | 11 | 11 | 11 |
| N50 scaffold size, nt | 3,500,734 | 3,339,398 | 3,270,165 |
| N90 scaffold size, nt | 2,134,864 | 1,697,083 | 1,880,400 |
| Scaffold max. nt | 4,859,873 | 5,366,244 | 4,830,258 |
| Scaffold min. nt | 9,086 | 38,780 | 280,724 |
| Number of scaffolds | 17 | 14 | 13 |
| N50 contig size, nt | 3,500,734 | 3,339,398 | 3,270,165 |
| N90 contig size, nt | 2,134,864 | 1,697,083 | 1,571,664 |
| GC content, % | 50.79 | 50.90 | 50.95 |
| Genes | 11,875 | 14,118 | 12,330 |
| BUSCO completeness v6.0.0 (basidiomycota_odb10), % | 98.6 | 99.4 | 99.3 |

To assess genome completeness and structural features of the PC9.15 assembly, we analyzed gene density, sequence coverage, and BUSCO scores. Gene density is consistent across 11 scaffolds, averaging approximately 41 genes per 100 kb, with an average sequencing depth of 71.8 reads per 10 kb (Fig 5A). Coverage is generally uniform across chromosomes, with localized increases on chromosomes 2 and 6 (Fig 5A), possibly due to repetitive sequences, such as transposable elements (TEs). Assembly completeness, assessed using the basidiomycota_odb10 database [35], yielded a BUSCO score of 99.4%, comparable to that of PC9 (98.6%) and PC15 (99.3%) [18], confirming the high quality of the PC9.15 assembly. Overall, these results confirm that the PC9.15 genome is well-assembled, complete, and suitable for further genomics analysis.

To evaluate genomic similarity and structural variation, we compared the PC9.15 genome to its parental strains, PC9 and PC15. Annotation of the PC9.15 genome using *funannotate* (v1.8.14) [36] identified 14,118 genes (S5 Table), higher than in PC9 (11,875 genes) [18] or PC15 (12,330 genes) [19–21]. To control for annotation inconsistencies arising from version differences, we re-annotated the PC9 genome using the same version of *funannotate* (v1.8.14), which yielded 12,003 genes. Therefore, the higher gene count observed in PC9.15 likely reflects a combination of improved assembly contiguity. To assess genome-wide relatedness, we used D-Genies to generate whole-genome dot-plot alignments between PC9.15 and its parental strains, PC9 and PC15 [37]. D-Genies summary statistics indicated that ~99.47% of the PC9 assembly aligned to PC9.15 at >75% nucleotide identity (Fig 5B), whereas only ~56.31% of the PC15 assembly aligned to PC9.15 under the same threshold. This demonstrates that PC9.15 is genetically closer to PC9, consistent with its congenic background. Comparative analysis revealed several chromosomal rearrangements in PC9.15 relative to PC9. Circos plot alignment showed that a segment from PC9 chromosome 4 had been translocated to PC9.15 chromosome 2, and a region from PC9 chromosome 7 translocated to and became inverted on PC9.15 chromosome 4 (Fig 5C). Additionally, we observed a unique region (segment A) on chromosome 4 of PC9.15 that did not align with the PC9 genome (Fig 5C), but partially matched chromosome 4 of PC15 (Fig 5D), indicating that this region retains PC15-derived sequence. The persistence of this segment may reflect suppressed recombination, potentially caused by repetitive sequences, structural rearrangements, or local chromosomal features that reduce crossover frequency, as observed in analogous recombination-suppressed regions [38].

Moreover, we annotated the PC9 mitochondrial genome, revealing a 65 kb circular molecule (S3 Fig). Given that PC9 and PC15 are descended from the strain N001, we expected their mitochondrial genomes to be identical. Indeed, our comparison showed 99.9% similarity between the mitochondrial genomes of PC9, PC15, and PC9.15 (S4 Fig and S6 Table).

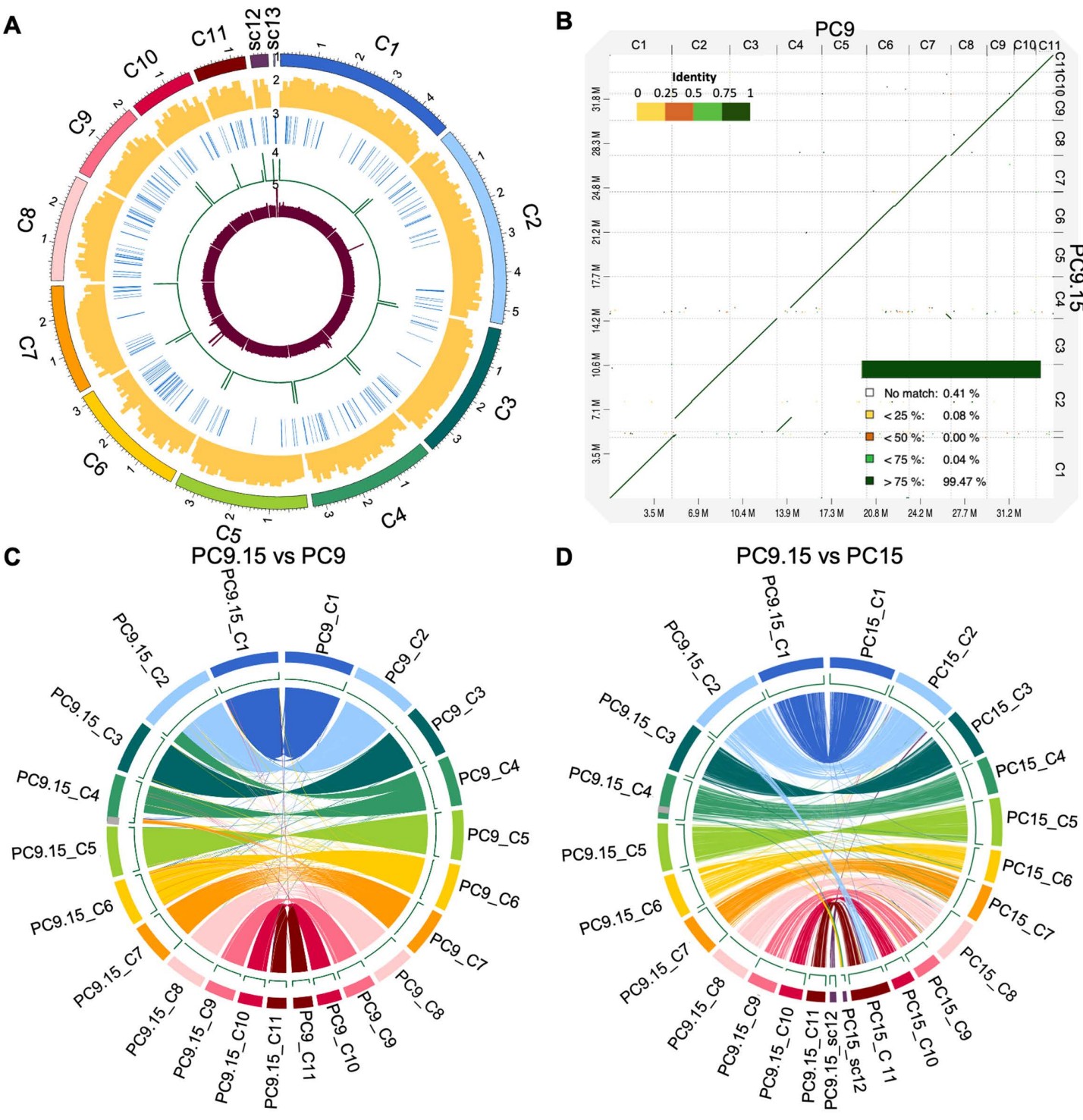

**Fig 5. Genome architecture of *P. ostreatus* congenic strain PC9.15 and comparative analysis of assembled genomes of PC9.15, PC9, and PC15. (A)** Tracks (outer to inner) represent the distribution of genomic features in our PC9.15 assembly: (1) size (in Mb) of PC9.15 scaffolds, with numbers prefixed by the letter "C" indicating the chromosome and "sc" indicating the scaffold; (2) gene density (along 100-kb sliding windows, ranging between 0 and 50 protein-coding genes); (3) distribution of TEs along the genome; (4) telomere repeat frequency (along 1-kb sliding windows, ranging between 10 and 30 repeats); and (5) depth of sequencing coverage (along 10-kb sliding windows, ranging between 0 and 150 depth). **(B)** Whole-genome dot-plot alignment of PC9.15 (query; y-axis) against PC9 (reference; x-axis) generated with D-Genies. Each dot represents an aligned segment,

colored by nucleotide identity as indicated in the scale. The summary values report the fraction of total query (PC9.15) sequence length that is unaligned ("No match") or aligns to the reference within each identity bin, including the fraction aligning at >75% identity (99.47%). **(C)** Circos plot showing regions of similarity shared by the PC9.15 (C1-C11) and PC9 (C1-C11) (identity > 95%, length > 10kb). **(D)** Circos plot showing regions of similarity shared by the PC9.15 (C1-C12) and PC15 (C1-C12) strains (identity > 95%, length > 10kb). The grey box represents segment A. Tracks (outer to inner) represent the distribution of genomic features in each assembly: (1) chromosome sizes (in Mb) of chromosomes, with numbers prefixed by the letter "C" indicating the chromosome and "sc" indicating the scaffold; (2) distribution of telomere repeats with 1-kb sliding windows, ranging between 10-30 repeats.

### Analysis of mating-type loci in the congenic strain PC9.15

To confirm the mating-type configuration of the congenic strain PC9.15, we analyzed the structure of its *matA* and *matB* loci and compared them to those of PC9 and PC15. As in parental strains, the *matA* and *matB* loci of PC9.15 are located on chromosome 2 and 9, respectively. The *matA* locus contains four *HD* genes (*HD1.2*, *HD1.3*, *HD2.2*, and *HD2.3*), and the *matB* locus includes five *STE3* genes (*STE3.7* to *STE3.11*) and six pheromone precursor genes (*PHB6* to *PHB11*), identical to PC15 (Fig 6A). These results confirm that PC9.15 carries the *A1B2* mating-type loci inherited from PC15.

To further identify the recombination sites around the mating-type loci, we expanded the sequence alignment to ~3 Mb for the *matA* locus and ~600 kb for the *matB* locus (Fig 6B). PC9.15 retained segments of the PC15 genomic background near both loci, with approximately 1.13 Mb around *matA* and 300 kb around *matB*. These retained regions raise the possibility of locally reduced recombination around the *mat* loci. Despite these regions of difference, the remainder of the PC9.15 genome is largely derived from PC9. Together, these data confirm the successful construction of a congenic strain that harbors the PC15 mating-type loci (*A1B2*) within the PC9 genetic background.

## Discussion

### Diversity and genomic organization of mating-type loci in *P. ostreatus*

In this study, we investigated the structure and diversity of *matA* and *matB* alleles in *P. ostreatus*. Given the diversity of mating-type loci across basidiomycetes, we compared our findings to other well-characterized systems. In *C. cinereus*, the *matB* locus is organized into three distinct gene groups, each containing a single pheromone receptor gene (*RCB*) and multiple pheromone genes (*PHB*) [25]. Similarly, phylogenetic analysis of Ste3 proteins in *P. ostreatus* revealed five distinct groups (Fig 2D). In most *P. ostreatus* strains, *PHB* genes are located near *STE3* genes, and the number of *PHB* genes is typically equal to or greater than the number of *STE3* genes. The frequent physical proximity of *PHB* and *STE3* gene, together with the higher copy number of *PHB* relative to *STE3*, indicates a characteristic genomic organization of the mating-type locus in *P. ostreatus*. This organization may allow *P. ostreatus* to fine-tune mating recognition through multiple pheromone-receptor combinations. Unlike the tightly clustered arrangement seen in *C. cinereus*, this dispersed structure likely reflects a different evolutionary path for generating mating-type diversity. In *P. ostreatus*, genes encoding the MAP kinase signaling cascade (e.g., *STE20*, *STE11*, *STE12*) are dispersed across multiple chromosomes rather than residing within a *MAT* locus as in *C. neoformans* [10]. Instead, the *matB* includes several reductases, transporters, and hypothetical proteins whose functions remain unclear but may still play a role in mating.

The tetrapolar mating-type system, with multiallelic *matA* and *matB* loci, promotes outcrossing in basidiomycetes [39]. For example, in *C. cinereus*, over 200 alleles per *MAT* locus yield a 99.5% outcrossing probability, whereas *Ustilago maydis*, with over 25 *b* alleles and only two *a* mating types, exhibits a lower outcrossing probability of 50% [40]. The diversity of mating-type alleles in *P. ostreatus* likely enhances outcrossing in natural populations. Our analysis identified 11 alleles at the *matA* locus and 12 alleles at the *matB* locus across 12 *P. ostreatus* haplotypes, suggesting the existence of at least 132 distinct mating types. These findings not only validate the multiallelic structure of *P. ostreatus* mating-type loci but also provide a detailed molecular characterization of allele diversity. Given this extensive allelic variation, additional *matA* and *matB* alleles likely remain unidentified in natural populations. Future studies on wild-isolated *P. ostreatus* strains will

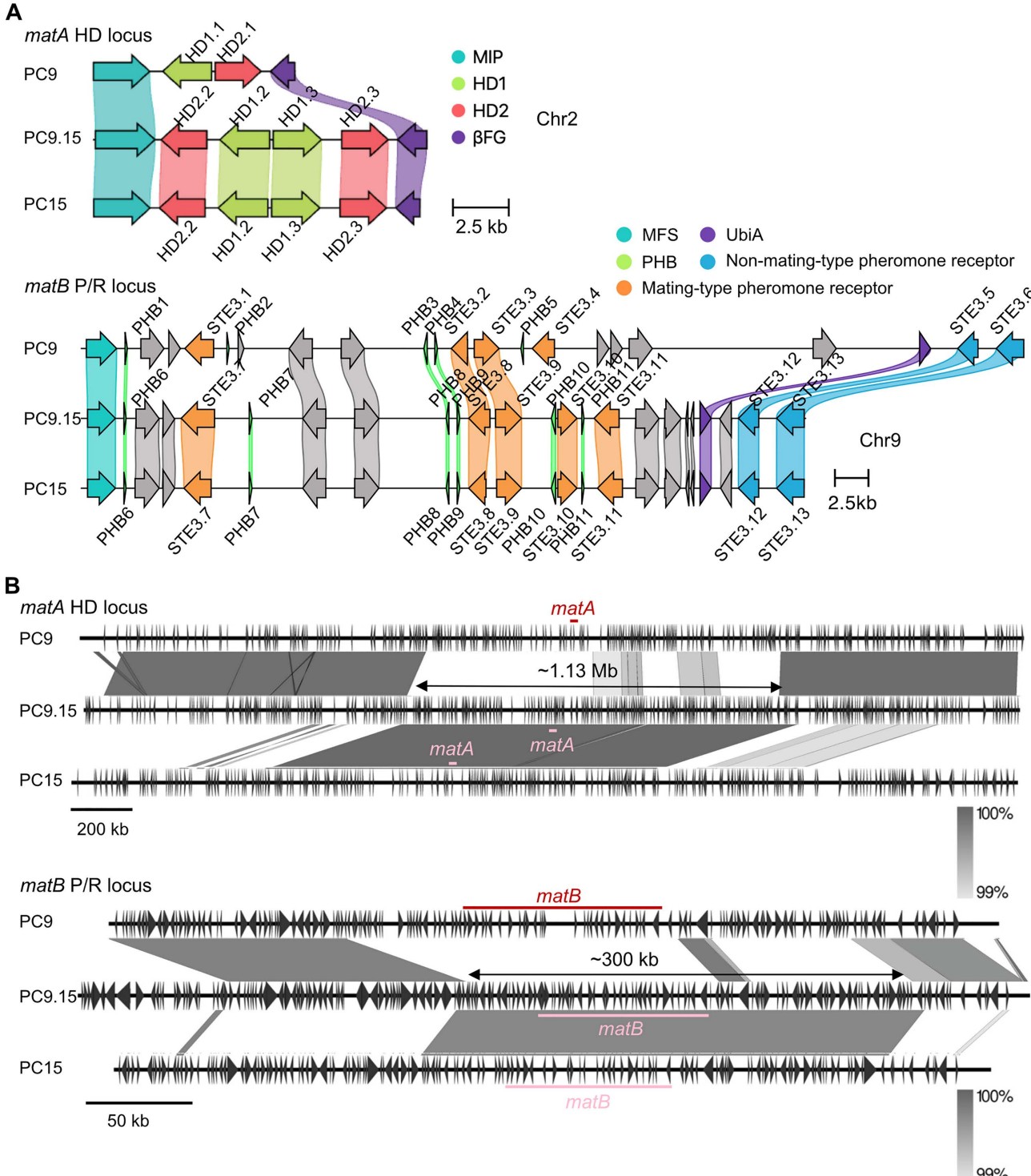

**Fig 6. Characterization of mating-type loci and associated genes in PC9.15.** Structures of the *matA* and *matB* loci (A) and the large-scale chromosomal (B) *matA* and *matB* loci compared across the PC9, PC9.15, and PC15 strains. **(A)** *MAT* locus alignments (amino acid-based) of strains PC9, PC9.15, and PC15 (clinker v0.0.31). Genes encoding proteins having more than 95% identical amino acid residues are linked. The color code for links and genes highlights sequence homology and grey genes have no similarity to other genes in the alignment. In the *matA* locus, homeodomain genes *HD1* and *HD2* are shown in green and pink, respectively; the conserved *MIP* and *βFG* are shown in teal and purple, respectively. In the *matB* locus,

mating-type specific pheromone receptor genes (*STE3*) are shown in orange (within *matB*) and blue (outside *matB*); pheromone genes (*PHB*) are in green; the conserved MFS and UbiA are shown in teal and purple, respectively; other genes are shown in grey. **(B)** Large-scale alignment of the *matA* and *matB* loci between PC9, PC9.15, and PC15. The *matA* and *matB* loci of PC9 is shown as red line; those of PC9.15 and PC15 are shown as pink lines. The shades of grey show nucleotide sequence identity. The figures were generated by Easyfig v2.2.2.

further enhance our understanding of the diversity and complexity of these loci and provide deeper insights into the evolution and regulation of the mating-type system in *P. ostreatus*.

## Expansion and allelic diversity of *MAT* loci across strains

Across twelve *P. ostreatus* haplotypes, we identified 11 *matA* alleles and 12 *matB* alleles, confirming extensive multiallelism at both *MAT* loci. We also observed significant expansion in the physical size of these loci, including one expanded *matA* locus (73 kb in the gfalt haplotype versus 6 kb in PC9) and three expanded *matB* loci (77–150 kb versus 50 kb in PC9). The expanded regions contain additional genes, notably a transposase, an intergrase/recombinase-like protein, and a retrotransposon Gag protein, suggesting that transposable elements (TEs) and retrotransposon activity may has contributed to the structural expansion. Similar TE-associated expansion of the *MAT* locus has been reported in *Cryptococcus*, where accumulation of transposons within the *MAT* region has contributed to its expansion [41]. Moreover, the transition from an ancestral tetrapolar to a bipolar mating system in *Cryptococcus* involved fusion of the previously unlinked *P/R* and *HD* loci via chromosomal translocations, resulting in an expanded contiguous *MAT* locus [42]. Together, these findings support the hypothesis that TE insertions, together with integrase/recombinase activity, create a genomic environment that promotes structural rearrangements while preserving mating-type diversity in *P. ostreatus*.

## Genomic rearrangement and recombination at the *MAT* loci

We observed regions surrounding both *MAT* loci, spanning ~1.13 Mb around *matA* and ~300 kb around *matB* lack recombination (Fig 6B). These regions are larger than the *MAT* loci themselves (*matA* ~ 10 kb; *matB* ~ 50 kb). Based on the *P. ostreatus* linkage map, the genome-wide average physical-to-genetic distance is estimated to be ~ 35 kb/cM [43], corresponding to ~ 32.1 cM for the 1.13 Mb *matA*-linked region and ~8.5 cM for the 300 kb *matB*-linked region. Thus, over 10 backcross generations, the probability of at least one recombinant arising within the ~ 1.13 Mb *matA* flanking region is relatively high (~96%), suggesting possibility of recombination suppression. On the other hand, the probability of at least one recombinant arising within the ~ 300 kb *matB* flanking region is 59%, leaving an ~ 41% chance that no recombinant occurs across this interval. Notably, recombination suppression around mating-type loci has been documented in other fungi; for example, *Schizothecium tetrasporum* exhibits suppression across a ~ 1.47 Mb region surrounding the mating-type locus [44].

Comparative analysis revealed several chromosomal rearrangements in the PC9.15 genome relative to PC9 and PC15. Chromosome translocations have been previously documented between PC9 and PC15 [18], so it is not surprising that the congenic strain also underwent large-scale rearrangements. For example, a portion of PC9 chromosome 4 (C4) has translocated to chromosome 11 (C11) in PC15 [18], and the same sequence (segment B) has translocated to chromosome 2 (C2) in PC9.15 (S5A and S5C Fig). Initially, we hypothesized that segment B originated from the centromeric region, which is often enriched in transposons and repetitive elements prone to recombination errors, as observed in *C. neoformans* [10]. However, further analysis revealed that segment B is present on both chromosome 2 (C2) and chromosome 4 (C4) in the PC9 genome, indicating that duplication events had occurred before the translocation was observed (S5B and S5D Fig). These findings suggest that the translocation of segment B likely resulted from recombination errors due to sequence similarity between duplicated segments.

Collectively, this study provides a detailed characterization of the *P. ostreatus MAT* loci in strains PC9 and PC15 and established PC9.15 as a congenic strain and a new genetic resource for precise genetic studies. We identified a multiallelic mating-type system with 11 *matA* and 12 *matB* alleles across twelve haplotypes. The PC9.15 congenic strain

established here offers a versatile tool for probing the genetic and evolutionary dynamics of mating-type loci and for conducting future functional studies in this species.

## Materials and methods

### Strains and culture conditions

*Pleurotus ostreatus* strains PC9 and PC15 differentiated from dikaryotic strain N001 [11], were used as the starting parental strains for genetic crosses. Spores were collected from a fruiting body of TWF713. Two monokaryon strains, SS5 and SS14, each originating from a single basidiospore, were established in the laboratory for analysis. Strains were maintained on yeast and malt extract with glucose (YMG) medium solidified with 1.5% (w/v) agar and incubated at 25°C.

### Crossing, fruiting conditions and isolation of meiotic progeny

To induce dikaryon formation, the mating-compatible monokaryotic strains PC9 and PC15 were co-cultured on YMG agar medium for two weeks. Agar blocks from the mating interface were then transferred into sterile bottles containing sawdust medium (100 g of sawdust, 30 g wheat bran, tap water to 150 mL). These bottles were then incubated at 25°C in darkness for ten days. To induce fruiting body formation and obtain F1 progeny, cultures were shifted to 15°C under a 12 hr light/12 hr dark cycle. Fruiting bodies appeared after approximately seven days and matured within 3–4 days. Spores (F1 progeny) were harvested in Millipore water and micromanipulated using a dissecting microscope, with each plate containing 18 basidiospores. To assess reproductive performance, we recorded the number of germinated progeny per plate. Individual progeny were cultured, and their mating types were determined by PCR using mating-type-specific primers (S4 Table).

### Congenic strain construction

PC9 was crossed with PC15, and single basidiospores were isolated. Three progeny carrying the PC15 mating type were selected and backcrossed to PC9. This process was repeated for a total of ten backcrosses to produce PC9.15, a congenic strain with mating loci derived from PC15 in the PC9 genetic background.

### Genome sequencing and assembly comparison

The congenic strain PC9.15 and the monokaryotic strains SS5, and SS14 were cultured on YMG agar at 25°C for 7 days. Mycelia were then transferred to 50 mL of YMG liquid medium and incubated at 25°C and shaken at 200 rpm for 4 days. Genomic DNA was extracted using the cetyltrimethylammonium bromide (CTAB) method and purified by extraction with chloroform and phenol-chloroform and precipitation with isopropanol and ethanol, followed by an AMPure XP (Beckman Coulter) purification step. Libraries were prepared using the genomic DNA by Ligation kit (SQK-LSK109), and Nanopore long-read sequencing was performed using a FLO-MIN106 R9 flow cell (Oxford Nanopore) for PC9.15 and a R10 version flow cell (Oxford Nanopore) for SS5 and SS14. This approach yielded ~0.6 M reads for PC9.15, with a mean length of 7,400 bp, resulting in ~128X genome coverage. For SS5 and SS14, sequencing produced ~0.76 M and ~0.65M reads, with mean read lengths of 11,981 bp and 10,348 bp, corresponding to ~202X and ~142X coverage, respectively. *De novo* assembly of Nanopore long reads was performed using NECAT [45].

For comparison of the PC9, PC15, and PC9.15 genomes, we used the dnadiff program from the MUMmer package [46], which computed sequence identity and generates alignment statistics between them.

### Genome annotation

Genome annotation for PC9.15, PC15, SS5, and SS14 was performed using *funannotate* (v1.8.1) [36] and the pipeline described at https://funannotate.readthedocs.io/en/latest/index.html. For PC9.15, we then used *funannotate train* with RNA-seq data and masked genome to generate preliminary gene models. RNA was extracted from the PC9.15 strain

and sequenced on an Illumina NextSeq 500 Mid Output 300 cycles using a paired-end 150 bp configuration (PE150). For strains PC15, SS5, SS14, CCMSSC00389 [29], CCMSSC03989 [30], gfPleOstr1.1 (haplotypes gf and gfalt) [31], and 595 [32], gene prediction was performed directly with *funannotate predict* after masking, without RNA-seq-based training. Functional annotation was performed using the *funannotate remote* option, which incorporated InterProScan and antiS-MASH to identify InterPro domains and secondary metabolite clusters, respectively. Finally, *funannotate annotate* was used to integrate functional annotations, including Pfam domains, clusters of orthologous groups (COGs), and EggNOG classifications. The full annotation of PC9.15 is provided in S5 Table.

The accession numbers of BioSample: SAMN05327826 (CCMSSC00389), SAMN09434509 (CCMSSC03989), SAMN23459969 (595), SAMEA8562045 (gfPleOstr1.1), and SAMEA8562045 (gfPleOstr1 alternate haplotype).

For STE3.2 in the PC9 genome, we manually curated the gene model by extending its 3' end to include RNA-seq-supported 3'UTR sequence that was missing from the previous annotation.

## Annotation of the PC9 mitochondrial genome

A putative large duplication in PC9 scaffold_14 was identified through self-alignment of the sequence and is suspected to be an artifact resulting from the circular nature of the mitochondrial DNA and overlapping regions in the assembled Nanopore reads. This hypothesis was confirmed by aligning the contig with the mitochondria genome of *P. ostreatus* P51 [47]. The duplicated region was manually removed, resulting in an assembled mitochondrial genome of PC9 that closely matched the size of the P51 mitochondria genome. Annotation was performed using MITOS2 [48].

## Genome and mating-type loci comparison

General genome assembly statistics, including scaffold length and N50, were calculated using the count_fasta_residues. pl script (https://github.com/SchwarzEM/ems_perl/blob/master/fasta/count_fasta_residues.pl). Assembly completeness was assessed with BUSCO v6.0.0 using the basidiomycota_odb10 database [35,49]. Repetitive elements were identified using a custom repeat library generated using a previously described pipeline [50], and analyzed with RepeatMasker [51], TransposonPSI [52], and LTRharvest [53]. The results were further analyzed using the one_code_to_find_them_all script [54]. Telomeric repeats were identified by scanning for the motif (TTAGGG)n in scaffold sequences [55].

Genome comparison between PC9 and PC9.15 was performed using Circos (v0.69.0) [56] and D-genies (Minimap v2.24) [37] to visualize synteny and structural differences. Mating-type loci from both genomes were aligned using Easyfig [57] or clinker [58], and protein-level comparison of HD1, HD2, Ste3, and Phb were performed with needleall [59]. Multiple sequence alignments of HD, Ste3, and Phb proteins were generated using MEGA11 with the MUSCLE algorithm [60]. A phylogenetic tree was constructed by using IQ-TREE v3.0.1 [61] and further visualized using iTOL v7 [62]. The HD1/HD2 and receptor protein sequences used for phylogenetic tree construction were obtained from *Schizophyllum commune*. The corresponding sequences were retrieved from NCBI GenBank. The GenBank accession numbers can be found in S7 Table.

Multiple sequence alignments of HD1, HD2, Ste3, and Phb proteins were generated with the T-coffee program [63], and alignment results were visualized with Jalview v2.11.5.0 [64].

## Cell wall staining and imaging

The fungal cultures were stained with SCRI Renaissance 2200 (SR2200), which labels fungal cell walls without affecting growth, and imaged using a LSM980 Airyscan 2 confocal microscope (Carl Zeiss).

## Statistics

Two-tailed unpaired Student's *t*-test was performed to determine the statistical difference between control and experimental samples in GraphPad Prism 9. $P < 0.05$ was considered significant; asterisks demonstrate statistical significance, as calculated by two-tailed unpaired Student's *t*-test (*$P < 0.05$, **$P < 0.01$, ***$P < 0.001$, and ****$P < 0.0001$).

## Supporting information

**S1 Fig. Protein sequence alignment of mating type-related proteins in *P. ostreatus*.** Alignment of (A) HD1, (B) HD2, (C and D) Ste3, and (E) Phb proteins from strains PC9 and PC15. Three levels of shading indicate degrees of sequence similarity: dark blue background with an asterisk (*) indicates identical amino acids, intermediate blue background with a colon (:) indicates conserved amino acids, and light blue with a dot (.) indicates semi-conserved amino acids. Grey and red shading indicate the homeodomain regions in HD proteins and the transmembrane (TM) helix regions in Ste3 proteins, respectively.
(TIFF)

**S2 Fig. Confirmation of mating-type loci in congenic strain PC9.15.** (A) PCR amplification of the *matA* and *matB* loci in PC9, PC15, and the congenic strain PC9.15 using mating-type specific primers: PC9-*matA*-F/R, PC9-*matB*-F/R, PC15-*matA*-F/R, and PC15-*matB*-F/R. (B) Germination rates of PC9 x PC15 and PC9 x PC9.15. Statistical significance was calculated using a two-tailed unpaired Student's *t*-test.
(TIFF)

**S3 Fig. Circular map of the mitochondrial genome of *P. ostreatus* PC9.** Tracks (outer to inner) show: (1) annotation of mitochondrial DNA-encoded genes: subunits of cytochrome c oxidase/complex IV (red), NADH dehydrogenase/complex I (yellow), ATP synthase/complex V (blue), apocytochrome b *cob* (green), ribosomal small subunit protein *rps3* (pink), DNA polymerase *dpo* (salmon) and DNA-directed RNA polymerase *rpo* (brown); (2) annotation of RNA genes: ribosomal RNA genes *rrnS* and *rrnL* (orange) and 25 transfer RNA (green).
(TIFF)

**S4 Fig. Comparative analysis of mitochondrial genomes of PC9, PC9.15, and PC15.** Dot-plot alignment generated by D-Genies minimap v2.24 showing the mitochondrial genomes of *P. ostreatus* strains: (A) PC9 (target) vs. PC15 (query), (B) PC9 (target) vs. PC9.15 (query), and (C) PC9.15 (target) vs. PC15 (query). Percentages indicate the fraction of the genome participating in alignments with >75% sequence identity.
(TIFF)

**S5 Fig. Translocation sites between the PC9 and PC9.15 genomes.** (A) Partial sequence alignment of PC9 chromosome 4 (PC9_C4), PC9.15 chromosome 2 (PC9.15_C2), and PC15 chromosome 11 (PC15_C11). (B) Partial sequence alignment of PC9 chromosome 4 (PC9_C4) and PC9 chromosome 2 (PC9_C2). (C and D) Circos plots showing regions of high similarity (identity > 95%, length > 10kb) between PC9 chromosome 4 (PC9_C4) and PC9.15 chromosome 2 (PC9.15_C2) (C) and PC9 chromosome 4 (PC9_C4) and PC9 chromosome 2 (PC9_C2) (D). Red boxes represent centromere region and yellow boxes represent segment B. The letter "C" indicates the chromosome.
(TIFF)

**S1 Table. *matA* genes and *matB* genes in PC9, PC15, and PC9.15 genomes.**
(XLSX)

**S2 Table. Ste3 proteins similarity used by needleall analysis.**
(XLSX)

**S3 Table. *matA* and *matB* genes in SS5, SS14, CCMSSC00389, CCMSSC03989, gfPleOstr1.1, gfPleOstr1.1 alternate haplotype, DSM11191, and 595 genomes.**
(XLSX)

**S4 Table. Primer sets for mating type detection.**
(XLSX)

**S5 Table. PC9.15 genome annotation table.**
(XLSX)

**S6 Table. The similarity of PC9, PC15, and PC9.15 mitochondrial genome.**
(XLSX)

**S7 Table. GenBank accession numbers of HD1/HD2 and pheromone receptors for phylogenetic analysis.**
(XLSX)

## Acknowledgments

We thank the Genomic Core of Institute of Molecular Biology, Academia Sinica, for Nanopore sequencing and the Bioinformatics Core of Institute of Molecular Biology, Academia Sinica, for genome assembly. We also thank Hillel Schwartz for comments and suggestions about this work.

## Author contributions

**Conceptualization:** Yi-Yun Lee, Yen-Ping Hsueh.

**Formal analysis:** Yi-Yun Lee, Guillermo Vidal-Diez de Ulzurrun.

**Funding acquisition:** Yen-Ping Hsueh.

**Investigation:** Yi-Yun Lee.

**Methodology:** Yi-Yun Lee, Yen-Ping Hsueh.

**Resources:** Yen-Ping Hsueh.

**Supervision:** Yen-Ping Hsueh.

**Validation:** Yi-Yun Lee.

**Visualization:** Yi-Yun Lee.

**Writing – original draft:** Yi-Yun Lee.

**Writing – review & editing:** Guillermo Vidal-Diez de Ulzurrun, Rebecca J. Tay, Yen-Ping Hsueh.

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
