## [Decision Letter · Decision Letter 0]

6 Feb 2026

PGENETICS-D-25-01267

Establishment of a Congenic Strain for the Oyster mushroom reveals the Structure and Evolution of Mating-Type Loci

PLOS Genetics

Dear Dr. Hsueh,

Thank you for submitting your manuscript to PLOS Genetics. I have had some difficulty recruiting reviewers, and in the interest of time, I have based my decision on the attached report. This is generally supportive, and recommends only minor revisions. I tehrfore ask you to submit a revised manuscript.

Please submit your revised manuscript within by Mar 08 2026 11:59PM. If you will need more time than this to complete your revisions, please reply to this message or contact the journal office at plosgenetics@plos.org. Please include the following items when submitting your revised manuscript:

We look forward to receiving your revised manuscript.

Kind regards,

Geraldine Butler

Section Editor

PLOS Genetics

Aimée Dudley

Editor-in-Chief

PLOS Genetics

Anne Goriely

Editor-in-Chief

PLOS Genetics

**Journal Requirements:**

3) We notice that your supplementary Figures are included in the manuscript file. Please remove them and upload them with the file type 'Supporting Information'. Please ensure that each Supporting Information file has a legend listed in the manuscript after the references list.

Potential Copyright Issues:

- Please confirm (a) that you are the photographer of Figures 2 and 3., or (b) provide written permission from the photographer to publish the photo(s) under our CC BY 4.0 license.

**Reviewers' comments:**

Reviewer's Responses to Questions

**Comments to the Authors:**

Reviewer #1: This is an interesting and well-structured manuscript that reports substantial progress towards facilitating molecular genetic analysis in the oyster mushroom Pleurotus ostreatus. This species is of interest as an experimental system, both because of its importance in agriculture as a food organism, and from a basic science perspective because of its ability to prey on nematodes.

The background to the work is that P. ostreatus was known to have a tetrapolar mating system involving two unlinked genetic loci (matA and matB) that have multiple (>10) alleles each. A pair of P. ostreatus lab strains, called PC9 and PC15, that can mate with each other was previously developed. However, PC9 and PC15 have quite different phenotypes (PC9 grows much faster) and there is substantial DNA sequence divergence between their genomes, which makes them unsuitable as a pair of parents for use in genetic mapping experiments. In the current manuscript, Lee et al have done two main things. (1) They identified and annotated the genes in the matA and matB loci, and examined the high sequence and structural diversity of these loci among ~10 isolates of P. ostreatus. (2) By repeated backcrossing, they introduced the matA and matB alleles from PC15 into the PC9 genetic background, to create a new strain called PC9.15; they show that PC9.15 and PC9 have similar phenotypes and can mate together. By genome sequencing, they show that the PC9.15 genome is almost identical to the PC9 genome except for islands of 300-1000 kb around the two introgressed mat loci, and some minor unplanned rearrangements. Therefore, PC9 and PC9.15 are suitable for use as parental backgrounds for future genetic analyses in P. ostreatus, such as mapping genes required for nematode killing. The manuscript reports an extensive body of work that has been done to a high standard. It makes a significant step forward for oyster mushroom, and for our understanding of mating-type locus population genetics and evolution in basidiomycete fungi.

I note that the manuscript mentions that the structure of the matA locus of P. ostreatus was previously reported by another group (Shnyreva and Shnyreva, 2023; ref. 13). I read this paper, and my opinion is that the new manuscript is far superior. The Shnyreva and Shnyreva paper did not examine the matB locus at all; their annotation overlooked one of the HD genes in the matA locus; they did not examine mat locus sequence/structure diversity with the species P. ostreatus; and they did not construct a pair of congenic mating strains. I think that Lee et al have given appropriate credit to it.

Major comment

L200-216: It’s not clear what criteria the authors use to differentiate between “distinct alleles” and “the same allele”. For example L202 says that “SS14 shares an identical matA allele with PC9 (allele A2)”. Does this mean identical nucleotide sequences of the whole matA locus, or just identical amino acid sequences of HD1.1 and HD2.1? Does each of the different HD1.x and HD2.x names in Figure 2 correspond to a different amino acid sequence?

Minor comments

There seems to be an error or contradiction regarding levels of sequence divergence in Ste3 proteins, between the data in Table S2 and the corresponding text on L175-178. Instead of using the terms “highly similar” and “considerably more divergent”, it would be more informative to state the actual percentage identities.

There also seem to be inconsistencies in the presence/absence of links connecting genes in Figure 1A. The legend says that links are drawn if sequence identity is >90%. However, Ste3.3 vs Ste3.9 is 96.9% identical (Table S2) but they are not linked. I wonder if some of the gray genes should also have links, e.g. the two gray genes between PHB1 and STE3.1 in PC9 seem to have two orthologs in PC15 (Figure 1A).

Assembly version and accession numbers need to be stated, both for the pre-existing genome sequences and for the new data, in Tables 1 and 2 and in some of the supplementary tables. For example, Table S3 contains lists of contig names and gene coordinates, but without accession numbers. On Line 137, please cite a publication and an assembly accession number for “the JGI genome”.

Figure 2A has two separate parts that are referred to at different places in the text, and so does Figure 2B. It would be better to call the 4 parts A-D.

L32, say that there were 10 rounds of backcrossing.

L145: Cite Figure 1A here.

L147: Do you mean “the matB locus in many basidiomycetes is bordered…” ?

L304: Delete the phrase “and enhanced performance of newer funnotate versions”, because there is a still 2000-gene difference between the PC9.15 annotation and the PC9 annotation when the same software version is used.

L306-309: I don’t think that the term “sequence similarity” is being used correctly in these sentences. I think that perhaps the numbers 99.47% and 56.31% refer to the percentage of the length of one genome that can be aligned to another genome, in alignments that have >75% sequence identity.

L346-359 and Figure 6: The section about the mitochondrial genome seems unnecessary and unrelated to the rest of the manuscript. I suggest deleting it except for the final sentence (L358), which can be moved to L320.

L372: I’m not sure why the authors suggest that mating regulation system in P. ostreatus is “more flexible”? Do you mean more flexible than in Coprinus cinereus? The P. ostreatus and C. cinereus systems seem to be very similar (they both have more PHB genes than STE3 genes), so I don’t understand why P. ostreatus is considered more flexible.

L415: Can we be certain that recombination is suppressed in these regions around the mat loci? I agree that the regions seem large, but are they larger than what is expected by chance, given the number of backcrosses and the recombination rate in P. ostreatus?

Figure 2B2: It would be helpful to highlight the Ste3 genes from PC9 and PC15 in red and yellow, as in Figure 1B.

L382 typo: cinereus

L608 typo: “(A)” is missing.

**Have all data underlying the figures and results presented in the manuscript been provided?**

Large-scale datasets should be made available via a public repository as described in the *PLOS Genetics*
data availability policy, and numerical data that underlies graphs or summary statistics should be provided in spreadsheet form as supporting information., and numerical data that underlies graphs or summary statistics should be provided in spreadsheet form as supporting information., and numerical data that underlies graphs or summary statistics should be provided in spreadsheet form as supporting information., and numerical data that underlies graphs or summary statistics should be provided in spreadsheet form as supporting information.

Reviewer #1: Yes

PLOS authors have the option to publish the peer review history of their article (what does this mean?). If published, this will include your full peer review and any attached files.). If published, this will include your full peer review and any attached files.). If published, this will include your full peer review and any attached files.). If published, this will include your full peer review and any attached files.

...

Reviewer #1: No

**Figure resubmission:**
---

## [Editor Report · Decision Letter 1]

27 Feb 2026

Dear Dr Hsueh,

We are pleased to inform you that your manuscript entitled "Establishment of a Congenic Strain for the Oyster Mushroom Reveals the Structure and Evolution of Mating-Type Loci" has been editorially accepted for publication in PLOS Genetics. Congratulations!

Yours sincerely,

Geraldine Butler

Section Editor

PLOS Genetics

Aimée Dudley

Editor-in-Chief

PLOS Genetics

Anne Goriely

Editor-in-Chief

PLOS Genetics

BlueSky: @plos.bsky.social

Comments from the reviewers (if applicable):

**Data Deposition**

If you have submitted a Research Article or Front Matter that has associated data that are not suitable for deposition in a subject-specific public repository (such as GenBank or ArrayExpress), one way to make that data available is to deposit it in the Dryad Digital Repository. As you may recall, we ask all authors to agree to make data available; this is one way to achieve that. A full list of recommended repositories can be found on our . As you may recall, we ask all authors to agree to make data available; this is one way to achieve that. A full list of recommended repositories can be found on our . As you may recall, we ask all authors to agree to make data available; this is one way to achieve that. A full list of recommended repositories can be found on our . As you may recall, we ask all authors to agree to make data available; this is one way to achieve that. A full list of recommended repositories can be found on our website....

http://datadryad.org/submit?journalID=pgenetics&manu=PGENETICS-D-25-01267R1

Additionally, please be aware that our data availability policy requires that all numerical data underlying display items are included with the submission, and you will need to provide this before we can formally accept your manuscript, if not already present. requires that all numerical data underlying display items are included with the submission, and you will need to provide this before we can formally accept your manuscript, if not already present. requires that all numerical data underlying display items are included with the submission, and you will need to provide this before we can formally accept your manuscript, if not already present. requires that all numerical data underlying display items are included with the submission, and you will need to provide this before we can formally accept your manuscript, if not already present.

**Press Queries**

If you or your institution will be preparing press materials for this manuscript, or if you need to know your paper's publication date for media purposes, please inform the journal staff as soon as possible so that your submission can be scheduled accordingly. Your manuscript will remain under a strict press embargo until the publication date and time. This means an early version of your manuscript will not be published ahead of your final version. PLOS Genetics may also choose to issue a press release for your article. If there's anything the journal should know or you'd like more information, please get in touch via plosgenetics@plos.org....

---

## [Editor Report · Acceptance letter]

PGENETICS-D-25-01267R1

Establishment of a Congenic Strain for the Oyster Mushroom Reveals the Structure and Evolution of Mating-Type Loci

Dear Dr Hsueh,

We are pleased to inform you that your manuscript entitled "Establishment of a Congenic Strain for the Oyster Mushroom Reveals the Structure and Evolution of Mating-Type Loci" has been formally accepted for publication in PLOS Genetics! Your manuscript is now with our production department and you will be notified of the publication date in due course.

With kind regards,

Judit Kozma

PLOS Genetics

On behalf of:
